# Effect of Supplementation with *Saccharomyces Boulardii* on Academic Examination Performance and Related Stress in Healthy Medical Students: A Randomized, Double-Blind, Placebo-Controlled Trial

**DOI:** 10.3390/nu12051469

**Published:** 2020-05-19

**Authors:** Michał Seweryn Karbownik, Joanna Kręczyńska, Paulina Kwarta, Magdalena Cybula, Anna Wiktorowska-Owczarek, Edward Kowalczyk, Tadeusz Pietras, Janusz Szemraj

**Affiliations:** 1Department of Pharmacology and Toxicology, Medical University of Lodz, Żeligowskiego 7/9, 90-752 Łódź, Poland; anna.wiktorowska-owczarek@umed.lodz.pl (A.W.-O.); edward.kowalczyk@umed.lodz.pl (E.K.); 2Department of Infectious Diseases and Hepatology, Medical University of Lodz, Kniaziewicza 1/5, 91-347 Łódź, Poland; joanna.kreczynska@gmail.com; 3Department of Pediatrics and Allergy, Copernicus Memorial Hospital in Łódź, Medical University of Lodz, Piłsudskiego 71, 90-329 Łódź, Poland; paulina_kwarta@poczta.onet.pl; 4Oklahoma Medical Research Foundation, Aging and Metabolism Program, 825 NE 13th St, Oklahoma City, OK 73104, USA; magdalena-cybula@omrf.org; 5Department of Clinical Pharmacology, Medical University of Lodz, Kopcińskiego 22, 90-153 Łódź, Poland; tadeusz.pietras@umed.lodz.pl; 6Department of Medical Biochemistry, Medical University in Lodz, Mazowiecka 6/8, 92-215 Łódź, Poland; janusz.szemraj@umed.lodz.pl

**Keywords:** *Saccharomyces boulardii*, dietary supplementation, probiotic, academic examination, stress, anxiety, salivary cortisol, salivary metanephrine, pulse rate

## Abstract

In recent years, bacterial probiotic dietary supplementation has emerged as a promising way to improve cognition and to alleviate stress and anxiety; however, yeast probiotics have not been tested. The aim of the present study was to determine whether 30-day supplementation with *Saccharomyces boulardii* enhances academic performance under stress and affects stress markers. The trial was retrospectively registered at clinicaltrials.gov (NCT03427515). Healthy medical students were randomized to supplement their diet with *Saccharomyces boulardii* CNCM I-1079 or placebo before sitting for an academic examination, which served as a model of stress. The grades of a final examination adjusted to subject knowledge tested in non-stressful conditions was used as a primary outcome measure. Psychometrically evaluated state anxiety, cortisol and metanephrine salivary levels, and pulse rate were tested at a non-stressful time point before the intervention as well as just before the stressor. Fifty enrolled participants (22.6 ± 1.4 years of age, 19 males) completed the trial in the *Saccharomyces* and placebo arms. Supplementation with *Saccharomyces* did not significantly modify examination performance or increase in state anxiety, salivary cortisol, and metanephrine. However, the intervention resulted in higher increase in pulse rate under stress as compared to placebo by 10.4 (95% CI 4.2–16.6) min^−1^ (*p* = 0.0018), and the effect positively correlated with increase in salivary metanephrine (Pearson’s r = 0.35, 95% CI 0.09–0.58, *p* = 0.012). An *intention-to-treat* analysis was in line with the *per-protocol* one. In conclusion, supplementation with *Saccharomyces boulardii* CNCM I-1079 appears largely ineffective in improving academic performance under stress and in alleviating some stress markers, but it seems to increase pulse rate under stress, which may hypothetically reflect enhanced sympathoadrenal activity.

## 1. Introduction

Cognitive abilities and academic achievements are of paramount importance to medical students as they shape professional competence, help succeed in future career, and contribute to overall life satisfaction [1,2]. One of the factors that potentially impairs cognitive efficiency and compromise academic performance is excessive stress. Threat response to a stressor, as found in a cold pressor test, may impair memory retrieval [3], which appears crucial to exam-passing rate [1]. Indeed, preexamination distress in medical students was found to impair cognitive function [4] and to predict poor academic performance [5,6,7]. Medical students tend to be more stressed than their peers, and academic examinations represent a significant source of this stress [5,8]. Such distress may not only impair examination performance but also deteriorate psychosomatic well-being; promote unhealthy behavior [5,7,9]; and lead to neuropsychiatric, cardiovascular, and other disorders [10,11]. The academic community should provide counselling on stress management and create a friendly learning environment to promote student well-being and to ensure the highest academic standards [5,8,11].

In recent years, probiotic dietary supplementation has emerged as a promising way to alleviate stress and to reduce anxiety and depression [12,13]. Such interventions have also been tested among medical students in a model of preexamination stress. Consumption of probiotic *Lactobacillus casei* strain Shirota YIT 9029 by healthy medical students exposed to examination stress resulted in an anxiolytic effect, as expressed by prevention of salivary cortisol increase but not by psychometric measures [14,15]. In addition, the intervention improved some indicators of sleep quality [16] and certain cold-like and gastrointestinal symptoms, among others [14,15], and affected serotonin metabolism [14]. Similarly, the administration of heat-inactivated *Lactobacillus gasseri* strain CP2305 to medical students preparing to take an examination improved sleep quality, prevented the increase of salivary cortisol, and affected stress-responsive microRNA expression [17]. Importantly, supplementation of a probiotic mixture containing *Lactobacillus* and *Bifidobacterium* exhibited stress-related increase in working memory [18], which could hypothetically enhance academic examination performance [1,19].

Although lactic acid and other bacteria tested as antianxiety and cognition-improving agents represent the vast majority of probiotic species, some yeast strains of *Saccharomyces* are also classified as probiotics [20]. *Saccharomyces boulardii*, the most investigated species of yeast probiotic, has a well-established therapeutic position in gastroenterology [21]. However, its effect on mental health and cognitive function, particularly in examination stress-exposed medical students, has received very little attention until now. Even so, the results documenting the antianxiety and neurocognitive effects of probiotic bacteria cannot be easily extrapolated to yeast probiotics due to their substantial biological differences [22,23].

The primary aim of the present study was to assess whether healthy medical students demonstrate better performance in academic examination when supplementing their diet for 30 days with a yeast probiotic strain *Saccharomyces boulardii* CNCM I-1079. The secondary aim was to evaluate the effects of this supplementation on preexamination psychological, physiological, and biochemical stress markers.

## 2. Materials and Methods

### 2.1. Study Design

The trial was performed in a unicenter, randomized, double-blind, placebo-controlled, parallel-group setting to investigate whether the intervention demonstrated superior results compared to placebo. The original protocol of the trial included, apart from placebo, two intervention arms: *Saccharomyces boulardii* and *Lactobacillus rhamnosus GG*. However, after trial completion, the *Lactobacillus* product was found to be degraded; the mean number of colony forming units (CFU) in a single dose was found to be 1.1 × 10^6^, i.e., 0.019% of that of declared (95% confidence intervals, CI 1.8 × 10^3^–7.2 × 10^8^). Therefore, the results of *Lactobacillus* intervention could be questionable. Consequently, the main analysis in the present article reports the effect of *Saccharomyces* only, whereas the effect of *Lactobacillus* is reported in Appendix A. The methods and detailed results of the internal quality control of the products used in the trial are reported in Appendix A.

### 2.2. Ethical Considerations

The Deans of the Faculty of Medicine and Faculty of Military Medicine as well as Bioethics Committee of the Medical University of Lodz approved the study (RNN/86/16/KE received on 19 April 2016). The study was carried out between 20 April and 20 June 2016. The study protocol was retrospectively registered in the U.S. National Institutes of Health on 09 February 2018 and is available at https://clinicaltrials.gov under accession number NCT03427515. Written informed consent was obtained from each participant at the entrance to the study after full explanation of its protocol. The study details were reported according to the CONsolidated Standards Of Reporting Trials (CONSORT) 2010 guidelines [24], and the checklist is presented in Appendix A.

### 2.3. Participants

Healthy third-year medical students were recruited on a voluntary basis. Volunteers considered for entry to the study had to meet the following eligibility criteria.

Inclusion criteria:being a third-year medical student of the Faculty of Medicine or Faculty of Military Medicine, Medical University of Lodz, Polandage 18–30 years

Exclusion criteria:formal inability to sit the first attempt of the final examination of Pharmacologychronic diseases: neurological, psychiatric, cardiological, gastroenterological, immunological, endocrine, or infectiousstate of immunosuppressionhistory of hospitalization (up to three months before entrance to the study)presence of central venous catheterparenteral nutritioncurrent pregnancy or intention to become pregnant within three months from the entrance to the studycurrent lactationallergic reaction (up to three months before entrance to the study)hypersensitivity to yeast, maltodextrins, potato starch, magnesium stearate, hypromellose, gelatin, glycerol, or titanium dioxidebody mass index over 30chronic medication use (up to three months before entrance to the study; “chronic” was defined from a frequency perspective as “at least 90 days a year on average”; pharmacological contraceptives were not considered “medication” and were allowed in the study)systemic antibacterial or antifungal medication use (up to three months before entrance to the study)overuse of alcohol (defined according to the Polish standards [25] as ≥20 g and ≥40 g per day for females and males, respectively) or psychoactive substances (up to three months before entrance to the study)tobacco smoking—more than 5 cigarettes (or equivalents) a day (up to three months before entrance to the study)pro- or prebiotic preparations intake (up to three months before entrance to the study)vegan or other atypical dietdoing professional or extreme sports

Any situation described in the exclusion criteria happening after volunteer enrollment resulted in exclusion from the study.

The required minimum sample size was determined with the use of G*Power version 3.1.9.2 [26]. The model of repeated measures analysis of variance (ANOVA) for within-between interaction was used (suitable for secondary outcome assessment). Statistical power was set to 0.8. As the estimation referred to no preliminary results, medium effect size f = 0.25 and default correlation among repeated measures r = 0.5 were assumed. The minimum total sample size was determined to be 42 in all three arms of the trial.

### 2.4. Intervention

Two probiotic dietary supplements commercially available in Poland were purchased:*LacidoEnter* capsules (Institut Rosell, Montreal, Canada; batch numbers HG09241 and HI17731; expiry date 01/2017 and 03/2017, respectively) containing lyophilized *Saccharomyces boulardii* CNCM I-1079 in a declared amount of 5 × 10^9^ CFU per dose,*Dicoflor 60* capsules (Dicofarm, Rome, Italy; batch numbers SM513 and SM514; expiry date of both batches 03/2017) containing *Lactobacillus rhamnosus GG* (ATCC 53103) in a declared amount of 6 × 10^9^ CFU per dose.

In order to obtain indistinguishable formulations, the content of the original capsules were transferred to the new gelatin capsules (ACG Associated Capsules, Maharashtra, India) and filled with a mixture of maltodextrins (Pepees, Łomża, Poland) *quantum satis*. Placebo products were obtained by filling the same capsules with the mixture of maltodextrins only. The formulations were prepared using a manual capsule filling device (Capsunorm; Eprus, Bielsko-Biała, Poland).

At the completion of the study, a sample of each product was subjected to internal quality control to determine the viability of probiotic organisms, and the obtained results were compared to the recommended minimum daily doses of the probiotics [21,27,28]. Details on the methods and results of this quality control are described in Appendix A.

### 2.5. Outcomes

The primary outcome of performance in academic examination was determined according to the number of correctly answered multiple choice questions in the computer-based final examination in basic medical pharmacology (hereinafter referred to as pharmacology). Selection of this examination was made on the basis of the results of a pre-study test performed before the trial in a group of fourth-year medical students (*n* = 91) who perceived this exam to be one of the most stressful: 11% of respondents described stress and anxiety experienced during this exam as “frightening”, 28% described it as “considerable”, 35% described it as “average”, 24% described it as “slight”, and 2% described it as “almost none”. The examination in pharmacology was carried out for third-year medical students in the summer semester. The study participants underwent the examination in three rounds held in two different days due to limited number of examination stations. Each round, another set of randomly chosen questions was included. The round was chosen by participants according to their own preferences.

A preexamination test in pharmacology was performed a day before the final examination to be used as a comparator for the results of the formal examination. Its result was not considered the formal academic assessment of the subject; therefore, it was assumed to reflect the actual preexamination subject knowledge with no effect of stress. This test was internet based (SurveyMonkey; SurveyMonkey, San Mateo, CA, USA) and consisted of 30 yes/no questions (one point for each correctly answered). The content of the test was assessed by two independent academic teachers from the Department of Pharmacology and Toxicology who were not involved in the trial as highly relevant for the examination in pharmacology. Two different versions of the test were used, each before a different day when the final examination was held.

The secondary outcome of stress markers included state anxiety, salivary cortisol, salivary metanephrine, and pulse rate. All the measures were assessed in both “basal” and “preexamination” settings, and the change between two time points was examined. While state anxiety, a distress indicator [11], is typically monitored in stress research [14,15] and salivary cortisol is a well-established biological marker of stress reaction [29], metanephrine, particularly salivary, is an uncommonly examined stress-responsive parameter. Metanephrine could be proposed as such to monitor sympathoadrenal manifestation, particularly to validate pulse rate changes [30,31]. Metanephrine is released to circulation as a relatively stable and inactive metabolite of catecholamine epinephrine [32], the major enhancer of heart rhythm [33], and the level of metanephrine correlates with respective catecholamine [34]. Unconjugated metanephrine is easily transported over the salivary gland membrane and is detectable in saliva; however, conflicting results regarding plasma-saliva metanephrine correlation have been published [35,36].

### 2.6. Procedure

The timeline of the study procedures is presented in Figure 1. Invitation to participate in the trial was addressed during a presentation in lectures and seminars in pharmacology as well as through social media of the Department of Pharmacology and Toxicology, Medical University of Lodz, which included brief summary of state-of-the-art knowledge about psychobiotics.

Preintervention period was initiated at the study inaugural session, during which enrollment of participants occurred and some basal data were collected. At the beginning of the session held in an academic assembly hall, full explanation of the trial protocol was provided. Evaluation for presence or absence of exclusion criteria was carried out by a medical doctor who explained each of the criteria collectively to all the volunteers and who was available for individual consultations on request. Written informed consent was obtained from the volunteers enrolled in the trial. The participants then drew paper sheets with printed numbers (hereinafter referred to as “participant code”) and were blinded to all the tests and biological samples throughout the trial with these codes. The name list of participant codes was prepared by participants with no supervision from investigators. A study participant who was the last to fill the list was asked to control it for legibility and completeness. The list was kept secured in a sealed envelope and was available to neither the investigators nor the outcome assessor. The list was intended in case the participant code was forgotten by the participant and to allow a person not involved in the trial to match the results of the final examination to participant codes after the study completion.

Afterwards, during the study inaugural session, basic sociodemographic data (sex; age; nationality; body mass index; smoking status; as well as one-item measures of perceived health status, eating a healthy diet, consumption of fermented products, and perceived economic status assessed in a 5-point Likert scale) were provided by the participants. Moreover, pulse rate was manually self-recorded for 30 s by palpation at the radial artery after at least 10-min obligatory rest in a sitting position. Instructions on how to measure the pulse rate as well as the counter that measured the required time were projected in the whiteboard. Finally, a set of psychometric questionnaires was applied, which included the following:Eating Attitudes Test-26 [37], Polish version [38], to measure the symptoms and concerns characteristic of eating disorders;Beck Depression Inventory (BDI) [39], Polish version [40], to measure severity of depression;Perceived Stress Scale-10 [41], Polish version [42], to measure perceived stress;State-Trait Anxiety Inventory (STAI) [43], Polish version [44], to assess both state and trait anxiety.

As this session was organized midsemester, which was considered a relatively non-stressful time period, state anxiety and pulse rate were assumed to be “basal” outcome measures, i.e., not affected by major stressful events.

Allocation of participants to the groups was carried out by a person not involved in the rest of the clinical trial to assure allocation concealment. The study employed a simple randomization procedure in a 1:1:1 ratio. The participant codes were entered to a spreadsheet in ascending order. Each code was assigned to a computer-generated pseudorandom integer between 1 and 3, each occurring with the same probability (Statistica 12.5; StatSoft, Tulsa, OK, USA). The 1-to-3 numbers were a priori attributed to the *Saccharomyces*, *Lactobacillus*, or placebo arms of the trial, respectively. The result of the group allocation was known to neither investigators nor participants to keep both groups blinded until completion of the study.

Participants were asked to collect their salivary samples at home in a non-stressful time in the preintervention period (to assess “basal” level of cortisol and metanephrine). Saliva was self-collected by chewing a polyethylene swab of Salivette Cortisol (Sarstedt, Numbrecht, Germany) in a seated position between 16:00 and 17:00 to eliminate the effect of diurnal fluctuations on measurands. Participants were informed to avoid intense physical activity; sexual intercourse; drinking alcohol, coffee, and other caffeine products; consuming blue cheese, nuts, pineapple, and bananas; and taking any medications unless prescribed by a doctor, particularly sympathomimetic drugs at the whole day of saliva collection. Additionally, from the midday, they were asked to avoid any physical activity, cigarette smoking, use of any other nicotine containing products, and drinking tea. They have been fasting at least 30 min before saliva collection. Thirty min before sampling, they also brushed their teeth without toothpaste, and 10 min before, they were asked to rinse their mouth with clean water, to prepare collection devices with indelibly written participant codes, and to sit calmly on a chair. The saliva samples were then stored in the domestic refrigerator and transported to the Department of Pharmacology and Toxicology, Medical University of Lodz on ice within 24 h from specimen collection, where they were immediately frozen at −20 °C until analysis.

While depositing a salivary sample in the department, participant received a product packaging (indistinguishable between different types of intervention) marked with their participant code. The packaging contained 30 capsules with an attached instruction to store it at room temperature in a dry and dark place throughout the study and to take one capsule a day, swallowed as a whole in the morning during or after the breakfast, washing down with a glass of water, for 30 days starting on the designated day. Together with the product, participants received “study diaries” to note any adverse event (AE) occurring at follow up.

At the end of the intervention period, a day before the final examination in pharmacology (day 29) between 16:00 and 17:00, participants collected a second sample of saliva at home (to assess the “preexamination” level of cortisol and metanephrine) following the same restrictions and with the same procedure as previously described, which was stored and transported under the same conditions. On that day, they also performed a preexamination test in pharmacology as described above.

On the day of the final examination (day 30), study participants were asked to come to the examination building 45 min before the exam started. They deposed their “preexamination” salivary samples collected a day before together with their study diaries. Afterwards, participants completed a “preexamination” state anxiety test with the suitable part of the STAI and self-recorded their “preexamination” pulse rate after at least 10-min obligatory rest according to the same protocol as previously described. After their final examination completion, study participants completed a final form providing information on the extent of adherence to the trial protocol and indicating their satisfaction with the study (assessed as 5-point Likert scale one-item measures). They were also asked a question: “which product, in your opinion, *Saccharomyces*, *Lactobacillus*, or placebo, did you take?”

### 2.7. Determination of Salivary Cortisol and Metanephrine

After being thawed, the salivary samples were centrifuged at 1000× *g* in 4 °C for 5 min, and the supernatants were transferred to new vials kept on ice. No apparent blood contamination was visible in any of the tested samples. The concentrations of cortisol and metanephrine were determined in salivary supernatants by an enzyme-linked immunosorbent assay (ELISA) with commercially available kits: Cortisol free in Saliva ELISA DES6611 (Demeditec Diagnostics, Kiel, Germany) and Metanephrine Plasma ELISA DEE8100 (Demeditec Diagnostics), respectively. Two technical replicates were run for each sample. BioTek EL ×800 microplate reader (BioTek, Winooski, VN, USA) was used to measure absorbance at 450 nm wavelength with a reference of 630 nm. A four-parameter logistic model was used to plot a calibration curve. The concentrations determined to be below the lower limit of quantification (LLOQ) were substituted by LLOQ/2 values [45]. Cortisol level was determined exactly according to the producer’s instructions. The metanephrine assay was modified and partially validated in terms of sample stability, protocol modification, and recovery to adjust to salivary samples [46,47]. Detailed methods and the results of the assay validation are presented in Appendix A.

### 2.8. Data Analysis

Descriptive statistics included means and standard deviations or absolute and relative frequencies, if not stated otherwise. Student’s *t*-test, asymptotic Mann Whitney *U* test, Pearson’s χ^2^ test, Fisher’s exact test, Kruskal–Wallis *H* test, and Pearson’s *r* correlation coefficient were used to compare basic sociodemographic and psychometric variables and in some ancillary analyses. The main analysis of outcome measures was performed *per protocol* (PP), i.e., for those participants who completed the trial (25 and 25 people in the group of *Saccharomyces* and placebo, respectively). Additionally, an *intention to treat* (ITT) sensitivity analysis was performed each time (31 and 29 people in respective groups).

The results of the final examination in pharmacology were recorded for all the enrolled participants, and the ITT analysis was straightforward. However, some the other outcome measures for dropout participants were missing and were filled in using a multiple imputation by chained equation (MICE) procedure under a *missing at random* assumption about the unobserved data [48]. The primary outcome of performance in academic examination was assessed using one-way block ANOVA with the examination set as a blocking factor. The results of the preexamination test in pharmacology held a day before the final examination were optionally included as an adjustment linear factor. Objective index of the final examination difficulty was assessed using Bruce’s Difficulty Coefficient (DC) [49], whereas internal consistency of preexamination tests was estimated with Kuder–Richardson Formula 20 (KR 20) [50].

Salivary cortisol and pulse rate data was log-transformed to bring the distribution closer to normal. The secondary outcomes of state anxiety, salivary cortisol, salivary metanephrine, and pulse rate were assessed using two-way repeated measures ANOVA. The between-subject factor was the group, whereas the within-subject factors were “basal” and “preexamination” time points. The two-way within–between interaction was evaluated each time. Presentation of the results of two-way interactions were preceded by the results of one-way repeated measures ANOVA performed separately in each study group to illustrate isolated effects of changes over time in PP analyses. Other general linear models were applied in ancillary analyses to adjust for factors of interest (stress reactivity, sex, self-reported consumption of fermented products, and basal pulse rate). *p*-values below 0.05 were considered statistically significant. The analysis was performed using STATISTICA 13.1 Software (StatSoft, Tulsa, OK, USA) and R Software version 3.6.1 (R Core Team 2019). The raw data was deposited in Mendeley Data repository (http://dx.doi.org/10.17632/jw3kppgytp.1).

## 3. Results

### 3.1. Flow of the Participants

Enrollment to the trial was carried out on 20 and 27 April 2016 in two equivalent inaugural sessions and resulted in the enrollment of 92 volunteers in total (60 further allocated to the groups of *Saccharomyces* and placebo). The preintervention period lasted for 21 to 33 days. Intervention started on 17 May (students of the Faculty of Medicine) and 22 May (students of the Faculty of Military Medicine). Participants took their capsules for 30 days until the final examination held on 15 and 20 June, respectively. The fraction of participants lost to follow up in all three study groups was 16/92, i.e., 17% (10/60, i.e., 17%, in the groups of *Saccharomyces* and placebo). Additionally, one participant from the group of placebo exhibited basal and preexamination salivary cortisol level that appeared outlying (20.4 ng/mL and 88.2 ng/mL, respectively), indicative of sample contamination or disease, and was removed from PP analyses. The study timeline is presented in Figure 1. A detailed analysis of dropouts in each group and the number of participants retained are presented in the study flow diagram (Figure 2).

### 3.2. Basal Characteristics of the Participants and Dropouts

The enrolled participants (*n* = 92) were 22.6 ± 1.3 years old; all of them were Polish (apart from a missing data on one subject); and 37 (40%) were male. Sociodemographic and basal psychometric data of all the participants are presented in Table 1. The results show no significant group differences between the characteristics.

Sixteen enrolled participants who dropped out from the trial differed in basal characteristics from those who were followed up. The dropouts had significantly higher perceived economic status (median: “very high”, 1st–3rd quartile: “quite high” to “very high” vs. median: “quite high”, 1st–3rd quartile: “quite high” to “quite high”, *Z* = 2.64, *p* = 0.0084) and significantly lower depressive characteristics as measured by BDI (median: 3, 1st–3rd quartile: 1–5 vs. median: 7, 1st–3rd quartile: 3–11, *Z* = −2.30, *p* = 0.021) accompanied with a trend to lower perceived stress (*Z* = −1.90, *p* = 0.058). Moreover, they achieved higher scores in final examination in pharmacology by 3.7 (95% CI 0.3–7.0) points (*t*(90) = 2.16, *p* = 0.033). The other characteristics were not significantly different.

### 3.3. Performance in Academic Examination

A PP analysis found no significant difference in final examination score between the *Saccharomyces* and placebo groups: 41.9 (95% CI 40.1–43.7) vs. 40.1 (95% CI 38.3–42.0), respectively (*F*(1,46) = 1.85, *p* = 0.18). After adjusting for the results of the preexamination test in pharmacology held a day before the final examination, the results remained similarly insignificant: 42.0 (95% CI 40.2–43.8) vs. 40.1 (95% CI 38.3–41.9), *F*(1,45) = 2.15, *p* = 0.15 (Figure 3). Also, no significant difference was observed between the *Saccharomyces* and placebo groups regarding the scores obtained in the preexamination test in pharmacology: 20.1 (95% CI: 18.6–21.6) vs. 20.7 (95% CI: 19.2–22.1), respectively (*F*(1,47) = 0.32, *p* = 0.57).

An ITT analysis resulted in a similar outcome but closer to statistical significance. The comparison of the final examination scores was 42.8 (95% CI 40.9–44.7) vs. 40.5 (95% CI 38.5–42.4), *F*(1,56) = 2.90, *p* = 0.094, while, after adjusting for the results of the preexamination test, was 42.8 (95% CI 41.0–44.6) vs. 40.4 (95% CI 38.5–42.3), *F*(1,55) = 3.27, *p* = 0.076. No significant difference between the *Saccharomyces* and placebo groups was found regarding preexamination test score: 21.0 (95% CI: 19.6–22.5) vs. 21.2 (95% CI 19.7–22.7), respectively (*F*(1,57) = 0.03, *p* = 0.86).

The difficulty of the final examination was average, indicated by the Bruce’s DC values of 7.3 and 10.0 for all of the students sitting the examination in the academic year from the Faculty of Medicine (*n* = 323) and Faculty of Military Medicine (*n* = 208), respectively. Preexamination tests exhibited moderate internal consistency as expressed by KR 20 0.57 and 0.66 for two different versions of the test. There were two missing values for the preexamination test (one in each study group), which were filled in using the MICE procedure before the primary outcome measure assessment.

### 3.4. State Anxiety

A PP analysis found state anxiety to significantly increase from “basal” to “preexamination” settings in both *Saccharomyces* (from 35.9 (95% CI 32.1–39.6) points to 52.7 (95% CI 48.5–56.9) points, *F*(1,24) = 39.11, *p* < 0.0001) and placebo groups (from 37.1 (95% CI 33.2–41.1) points to 53.1 (95% CI 49.1–57.1) points, *F*(1,24) = 86.69, *p* < 0.0001. There was no significant difference between the extent of increase in state anxiety between the *Saccharomyces* and placebo groups: 16.8 (95% CI 11.3–22.3) points vs. 16.0 (95% CI 12.4–19.5) points, respectively (two-way interaction, *F*(1,48) = 0.07, *p* = 0.79) (Figure 4A).

An ITT analysis resulted in a similar outcome. No significant difference was found between the extent of increase in state anxiety between the *Saccharomyces* and placebo groups: 16.2 (95% CI 11.0–21.4) points vs. 16.9 (95% CI 13.3–20.4) points, respectively (two-way interaction, *F*(1,58) = 0.05, *p* = 0.83).

### 3.5. Salivary Cortisol

A PP analysis found salivary cortisol level to significantly increase from “basal” to “preexamination” settings in both *Saccharomyces* (from 2.22 (95% CI 1.59–3.09) ng/mL to 3.20 (95% CI 2.51–4.08) ng/mL, *F*(1,24) = 5.29, *p* = 0.030) and placebo groups (from 2.13 (95% CI 1.66–2.74) ng/mL to 3.06 (95% CI 2.24–4.18) ng/mL, *F*(1,24) = 5.03, *p* = 0.034). There was no significant difference between the extent of increase in salivary cortisol level between the *Saccharomyces* and placebo groups: 0.98 (95% CI 0.10–1.86) ng/mL vs. 0.93 (95% CI 0.07–1.78) ng/mL, respectively (two-way interaction, *F* (1,48) < 0.01, *p* = 0.98) (Figure 4B).

An ITT analysis resulted in a similar outcome. No significant difference was found between the extent of increase in salivary cortisol level between the *Saccharomyces* and placebo groups: 1.55 (95% CI 0.66–2.44) ng/mL vs. 1.04 (95% CI 0.11–1.98) ng/mL, respectively (two-way interaction, *F* (1,58) = 0.97, *p* = 0.33).

While measuring salivary cortisol level with ELISA, the experimentally set intra- and inter-assay coefficient of variations (CVs) were 7.0% and 10.6%, respectively. LLOQ of cortisol reported by the producer was 0.024 ng/mL, and none of the analyzed salivary samples fell below this value.

### 3.6. Salivary Metanephrine

A PP analysis did not indicate any significant increase in salivary metanephrine level from “basal” to “preexamination” settings in the *Saccharomyces* (from 29.7 (95% CI 23.0–36.4) pg/mL to 38.7 (95% CI 27.4–49.9) pg/mL, *F*(1,24) = 3.86, *p* = 0.061) or placebo groups (from 33.9 (95% CI 26.3–41.4) pg/mL to 33.5 (95% CI 25.0-42.0), *F*(1,24) = 0.02, *p* = 0.90). There was no significant difference between the extent of change in salivary metanephrine level between the *Saccharomyces* and placebo groups: 9.0 (95% CI −0.5 to 18.5) pg/mL vs. −0.4 (95% CI −7.2 to 6.4) pg/mL, respectively (two-way interaction, *F* (1,48) = 2.78, *p* = 0.10) (Figure 4C).

An ITT analysis resulted in a similar outcome. No significant difference was found between *Saccharomyces* and placebo groups regarding the change in salivary metanephrine level: 11.0 (95% CI 2.5 to 19.5) pg/mL vs. 1.5 (95% CI −4.8 to 7.8) pg/mL, respectively (two-way interaction, *F* (1,58) = 3.32, *p* = 0.073).

While measuring salivary metanephrine level with ELISA, the experimentally set intra- and inter-assay CVs were 12.3% and 17.8%, respectively, being consistent with assay characteristic reported by the producer. The adjusted LLOQ (see Appendix A) was 6.6 pg/mL, and five (5%) of the analyzed salivary samples fell below this value.

### 3.7. Pulse Rate

Pulse rate significantly increased from “basal” to “preexamination” time points in both *Saccharomyces* (from 68.9 (95% CI 65.1–72.9) min^−1^ to 90.3 (95% CI 84.9–96.1) min^−1^, *F*(1,24) = 74.93, *p* < 0.0001) and placebo groups (from 72.9 (95% CI 68.4–77.7) min^−1^ to 83.9 (95% CI 78.2–90.0) min^−1^, *F*(1,24) = 35.22, *p* < 0.0001) in a PP analysis. However, the extent of pulse rate increase was significantly higher in the group of *Saccharomyces* than placebo: 21.4 (95% CI 16.3–26.5) min^−1^ vs. 11.0 (95% CI 7.2–14.9) min^−1^, respectively (two-way interaction, *F*(1,48) = 10.91, *p* = 0.0018) (Figure 4D).

An ITT analysis confirmed this result. The extent of pulse rate increase was significantly higher in the group of *Saccharomyces* than placebo: 23.2 (95% CI 18.1–28.2) min^−1^ vs. 11.8 (95% CI 8.1–15.5) min^−1^, respectively (two-way interaction, *F*(1,58) = 13.20, *p* = 0.0006).

### 3.8. Ancillary Analyses

To explore the relationship between the reported effects, Pearson’s correlation coefficients were determined in the polled study groups. The extent of “basal” to “preexamination” increase in pulse rate was predicted by both the increase in salivary metanephrine (r = 0.35, *p* = 0.012) and the increase in state anxiety (r = 0.29, *p* = 0.040) in a PP analysis (Table 2). An ITT analysis led to the same statistical conclusion.

To further explore whether the reported effects are consistent between individuals with different levels of some covariates, PP and ITT analyses (two-way within–between ANOVA interactions) were repeated with adjustment to stress reactivity (defined as a difference in “preexamination” and “basal” state anxiety), sex, self-reported consumption of fermented products, and basal pulse rate. The results were consistent with the unadjusted analyses presented above.

Moreover, the three-way interaction of the within-subjects factor, the study group, and a covariate was assessed in PP and ITT analyses to evaluate whether the effect of *Saccharomyces* differs between individuals with various levels of a covariate. The majority of the results implicated no influence of a covariate on the effects of interest. The only difference was in relation to consumption of fermented products and basal pulse rate: the individuals who reported consuming more fermented products (or having lower basal pulse rate) presented higher increase in pulse rate under stress in response to *Saccharomyces* supplementation (three-way interaction, *F*(1,46) = 7.40, *p* = 0.0092 in a PP analysis for consumption of fermented products and *F*(1,56) = 4.67, *p* = 0.035 in an ITT analysis for basal pulse rate). Moderating the effect of basal pulse rate may reflect the same phenomenon as represented by consumption of fermented products. The details of these analyses are reported in Appendix A and Appendix A.

Analysis of the post-examination survey revealed the following results. Participants who completed the trial admitted good adherence to the study protocol with no significant difference between the study groups (median, 1st–3rd quartiles: 4, 4–5 vs. 4, 4–5 vs. 4.5, 4–5, in the groups of *Saccharomyces*, *Lactobacillus*, and placebo, respectively; Kruskal–Wallis *H* test, χ^2^(2) = 0.47, *p* = 0.79). Study participants could not identify the product that they had taken (Pearson’s χ^2^(4) = 5.70, *p* = 0.22, Table 3), which confirmed appropriate blinding of the participants and indistinguishability of the product formulations. A total of 57 participants (74%) assessed the trial as “very well” prepared and conducted, whereas the rest of them returned “quite well”.

To assess sample representativeness, some characteristics of the enrolled volunteers (*n* = 92) were compared to that of their classmates who decided to not participate in the trial or who met any exclusion criteria (*n* = 439). The enrollment rate was not significantly different between the students from Faculty of Medicine and those of the Faculty of Military Medicine (odds ratio 1.50, 95% CI 0.93–2.42, χ^2^(1) = 2.73, *p* = 0.098). The odds of the women enrolled to the trial was also not significant (1.06, 95% CI 0.67–1.68, χ^2^(1) = 0.07, *p* = 0.79). The scores in the final examination in pharmacology of enrolled students was not significantly different from that of those who were not enrolled: two-way block ANOVA with “faculty” as a blocking factor, mean score difference 1.1, 95% CI −0.4 to 2.7, *F*(1,528) = 2.10, *p* = 0.15).

The CONSORT 2010 checklist is presented in Appendix A. The effect of intervention with *Lactobacillus* product is reported in Appendix A. The results of the internal quality control of the products used in the trial are reported in Appendix A. The results of the partial validation of metanephrine assay for the use with saliva samples are reported in Appendix A.

### 3.9. Harms

During 30-day follow-up, two out of 82 participants who started the intervention (2%) reported an infection that required antibiotic use: one in a group of *Saccharomyces* and one in *Lactobacillus* (Fisher’s exact test, *p* = 1.0). These participants were excluded from the trial. According to the study diaries of the participants who completed the trial, 26 participants (34%) reported any AE in the 30-day follow-up period. There was no significant difference between the incidence of any AE between the study groups (Pearson’s χ^2^(2) = 0.05, *p* = 0.97). The reported AEs were as follows:cold—experienced by 11 participants (14%) for the median number of days of 3 (1st–3rd quartile: 2–5) in 30-day follow-up; no significant difference in the number of days with cold were reported between the study groups (Kruskal–Wallis *H* test, χ^2^(2) = 0.82, *p* = 0.66);fever—experienced by two participants (3%) for the median number of days of 1.5 (1st–3rd quartile: 1–2) in 30-day follow-up; both cases were in the *Lactobacillus* group; however, the difference in the number of days with cold were found statistically insignificant between the study groups (Kruskal–Wallis *H* test, χ^2^(2) = 3.97, *p* = 0.14);gastrointestinal symptoms—experienced by 22 participants (29%) for the median number of days of 2 (1st–3rd quartile: 1–3) in 30-day follow-up; no significant difference in the number of days with cold were reported between the study groups (Kruskal–Wallis *H* test, χ^2^(2) = 0.13, *p* = 0.94).

No other AE were reported.

## 4. Discussion

There is an accumulating volume of evidence indicating that bacterial probiotics alleviate anxiety [12,13]. This is also the case under academic examination stress [14,15,17] and may translate to improved cognition [18]. However, fungal probiotic species have been neglected in research on stress, anxiety, and cognitive function. The current report complements these shortcomings. It provides largely negative results as *Saccharomyces boulardii* supplementation was found neither to significantly affect academic performance under stress nor to relieve state anxiety or to prevent increase of salivary cortisol and metanephrine.

Such negative results are not uncommon in scientific literature. Not only individual clinical trials but also some systematic reviews found no significant or inconsistent effect of probiotic supplementation on anxiety [51,52] or any psychological outcome [53] in humans. Apart from methodological issues, this could result from application of diverse strains of probiotics [52]. In the present study, a probiotic strain was applied that is totally distinct from commonly used lactic acid and other bacterial probiotics, as it represents the eukaryote empire, fungal kingdom. Fungal probiotics significantly differ from bacterial ones. Their cell sizes are much larger, cell walls are composed of different elements, and they are less able to colonize the gut. Importantly, both the groups differ in mechanism of action and beneficial properties [22,23]. Because of these reasons, no one may legitimately extrapolate the results of clinical trials involving bacterial probiotics to fungal ones. In this light, the current negative results of the *Saccharomyces boulardii* supplementation on academic performance under stress and some stress markers offer a novel portion of evidence.

Although the present report appears essentially discouraging, it provides some evidence of sympathoadrenal stimulation following the supplementation. The intervention was found to accelerate pulse rate under stress. The extent of this acceleration was associated with increase in salivary methanephrine, which may validate the finding to some extent [30,31]. No significant effect of the supplementation on salivary metanephrine itself was observed; however, it was close to significance level. Moreover, the stability of the effect of pulse rate increase between participants of different stress reactivity [54], sexes [55], self-reported consumption of fermented products [22,23], and basal pulse rate may further support the robustness of the obtained results.

Soares et al. [56] supplemented rats with *Saccharomyces boulardii* for 10 days; their results indicate that the probiotic increases maximal rate of oxygen consumption following fatiguing treadmill running. The authors provide a hypothesis that such a performance-enhancing effect may be determined by the “ability of the cardiovascular system to specifically increase coronary and skeletal muscle blood flow”. If such a hemodynamic effect is secondary to cardiac output, it may be in line with the present findings of *Saccharomyces*-mediated increase in pulse rate under stress.

The attempts to resolve the mechanism underlying the effect of pulse rate increase reported in the current research may only be speculative and inconclusive as little data links *Saccharomyces boulardii* with sympathoadrenal response and Soares et al. [56] did not formally investigate the mechanism resulting in the effect they observed. Some proposals of the mechanism follow. Firstly, bacterial probiotics have been suggested to exert antidepressant and anxiolytic effects through the afferent vagus nerve [57]. The same pathway was found to mediate tachycardia in response to esophagus distention in rats [58]. However, it is not known whether *Saccharomyces* stimulates the gut–brain axis through the vagus nerve and how much this pathway is relevant to the observed effect of pulse rate increase. Secondly, it was reported that *Saccharomyces* exhibits a comprehensive anti-inflammatory effect by inhibition of pro-inflammatory cytokines production. This is mediated through nuclear factor κB, mitogen-activated protein kinases ERK1/2 or p38, as well as peroxisome proliferator-activated receptor gamma [59]. Some cross-talk between inflammatory response and the autonomic nervous system has been described [60] and could hypothetically contribute to the observed effect. Thirdly, it was found that *Saccharomyces* directly produces catecholamines [61]. It is unlikely, however, that such catecholamines increase pulse rates as they are poorly orally absorbed [62]. On the other hand, catecholamines released in the gastrointestinal tract may stimulate the growth of other microorganisms contributing to microbiota alterations [63]. Fourthly, West and al. [64] stressed mice before euthanizing them and perfusing their intestine tissues ex vivo with *Saccharomyces boulardii* suspension. The authors observed that the probiotic application normalized acute stress-related gut dysmotility and attributed the effect to the local action of yeast on the intestine tissue. Although the present study reports no significant effect of the *Saccharomyces* on gut motility of healthy volunteers under stress, a similar mechanism mediating the pulse rate response cannot be excluded [65]. Additionally, the need for an interplay between supplemented *Saccharomyces* and other microbes may be suggested [22,23] as the effect of pulse rate increase was particularly highlighted in consumers of fermented products. Finally, type I error should be simply considered. This is particularly the case as the self-reported manual measure of pulse rate suffers from limited validity. Although it was performed by third-year medical students who were taught how to perform the measurement many times during their education, accuracy of such methods under stress may be substantially impaired [66,67]. No validation of self-monitored manual pulse rate measurement under stress against more valid electronic or optical methods was carried out.

If the effect of *Saccharomyces*-mediated increase in pulse rate under stress is true-positive, one should consider whether it appears beneficial or harmful. The current study provides no response to this question. In line with Soares et al. [56], increase in pulse rate might help perform better during physical exercise. Conversely, in the research by Papalini et al. [18], probiotic-induced stress-related cognition enhance was accompanied with no sympathoadrenal response. Additionally, the increase in pulse was linked to increase in state anxiety as reported in the current study, which may suggest a detrimental psychological effect. Moreover, the pulse-increasing effect should raise concerns as it constitutes a cardiovascular risk factor [68].

The present study examined only a particular probiotic strain of *Saccharomyces boulardii* (CNCM I-1079) [69], and the results cannot be generalized to other strains such as CNCM I-745 [70]. It was postulated that even distinct technological preparations of the same probiotic strain might affect clinical outcome [71]. Regarding the application of academic examination in stress research, Stowell [49] raised some methodological doubts. The present study, however, acceptably fulfills his criteria for an appropriate model of stress. Regarding the sample representativeness, although several exclusion criteria existed in the trial, the sample was consistent with the third-year medical student population of the Medical University of Lodz. Moreover, the study sample presented trait anxiety [72] and depressive features [73] comparable to that of other examined samples of medical students, suggesting its representativeness of the population of medical students. Taking into account some dissimilarity of medical students from the general population of young adults [8,74], no further attempts of generalization are recommended.

Apart from generalizability issues, the present study has some substantial limitations that warrant mention while interpreting the results. Firstly, the methods used to assess sympathoadrenal activity are modestly representative. Self-reported pulse rate may not be accurate enough as discussed above. No tests other than salivary metanephrine were performed to confirm sympathoadrenal effect. Further studies would require other and more objective cardiovascular measures such as electrocardiogram, heart rate variability, blood pressure monitoring, metanephrine together with normetanephrine assessment, and possibly other time-points or within-examination monitoring. Secondly, the performed analyses may be underpowered to detect some of the effects as probability values of some statistical tests consistently approach significance level. At this time, however, no conclusion can be drawn from *p*-values not exceeding significance level. Thirdly, although the self-reported trial protocol adherence was monitored, neither any adherence tracking devices were used nor the stool specimens were evaluated for presence of supplemented yeast. Fourthly, it cannot be expected that the postulated effect lasts long following supplementation completion as *Saccharomyces boulardii* has the ability to survive in healthy human intestines no more than a few days [70]. Fifthly, the results should not be interpreted as a stimulus for pharmacological management of stress to not induce overmedicalization [75]. Instead, effective coping strategies together with psychological interventions should be encouraged. Sixthly, failure to include an acceptable quality comparator of *Lactobacillus* intervention leaves the obtained results with no relation to a well-investigated anxiolytic effect of bacterial probiotics [12,13]. Lastly, the trial was registered in a public register after its completion, which may be regarded as holding potential publication bias [76]. The authors were not aware of the need of preregistration at that time, as they confused the national guidelines on registration of clinical trials of medicinal products only [77] with the requirements of the International Committee of Medical Journal Editors [78]. The authors attest that, apart from this important issue, they made every effort to comply with the highest methodological standards.

## 5. Conclusions

The study reveals for the first time the effect of 30-day supplementation with a fungal probiotic *Saccharomyces boulardii* CNCM I-1079 on academic examination performance and related stress in healthy medical students. The supplementation has significant effect on neither performance in academic examination nor state anxiety, salivary cortisol, and metanephrine. However, the supplementation appears to increase pulse rate under stress, which may hypothetically reflect enhanced sympathoadrenal activity. The results of pulse rate increase should be regarded with caution due to limited validity of the measure. The findings require confirmation and exploration at clinical, preclinical, and in vitro levels, applying more accurate methods. The study has the potential to drive research forward particularly in the neglected field of pro-cognitive and antianxiety properties of fungal probiotics.

## Figures and Tables

**Figure 1 nutrients-12-01469-f001:**
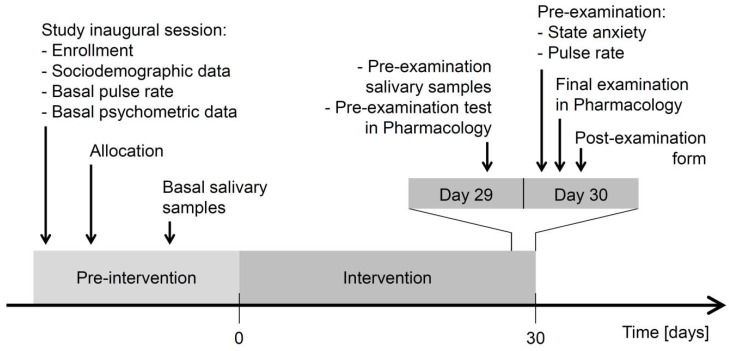
Timeline of the study procedures.

**Figure 2 nutrients-12-01469-f002:**
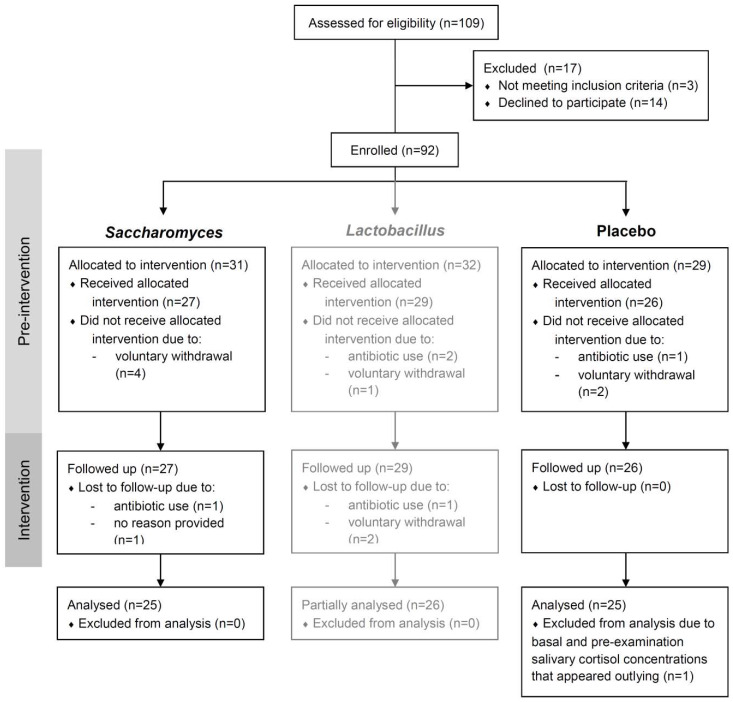
Study flow diagram.

**Figure 3 nutrients-12-01469-f003:**
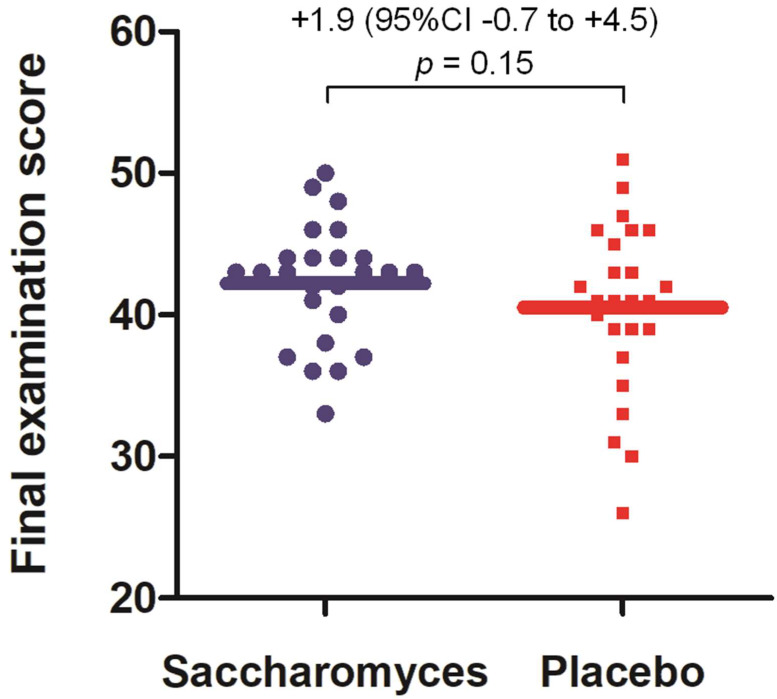
Primary outcome measure: final examination scores. A *per protocol* analysis is presented. Individual data points are marked. The mean scores are represented by horizontal lines. The effect sizes of the difference in the scores together with a *p*-value for the comparison are reported above the graph. The analysis is presented after adjusting for the results of the preexamination test in pharmacology held a day before the final examination.

**Figure 4 nutrients-12-01469-f004:**
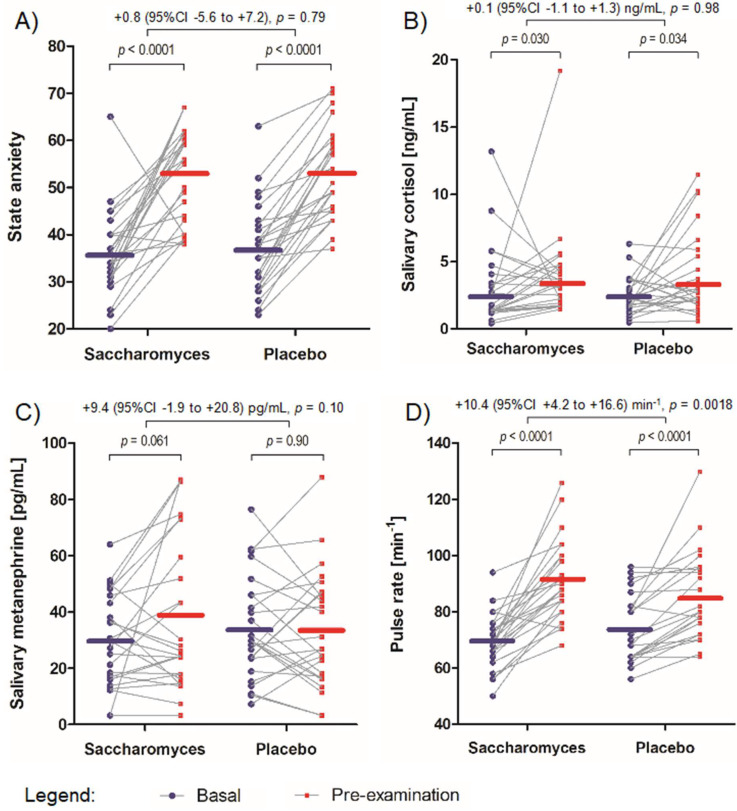
Secondary outcome measures: Individual data points are marked and linked with a grey line between “basal” and “preexamination” time points. The mean scores are represented by horizontal lines for both “basal” and “preexamination” settings. The effect size of the differences between the increase in the measures together with *p*-values for the comparisons are reported above the graph. (**A**) The extent of increase in state anxiety between the study groups. (**B**) The extent of increase in salivary cortisol level between the study groups. (**C**) The extent of increase in salivary metanephrine level between the study groups. (**D**) The extent of increase in pulse rate between the study groups.

**Table 1 nutrients-12-01469-t001:** Basal sociodemographic, psychometric, and laboratory characteristics of the participants enrolled to the study: The results include means and standard deviations or absolute and relative frequencies, if not stated otherwise.

Variable	Arm of the Trial			Test Statistics and *p*-Values for the Comparison of	
	*Saccharomyces* (*n* = 31)	*Lactobacillus* (*n* = 32)	Placebo (*n* = 29)	*Saccharomyces* vs. Placebo	*Lactobacillus* vs. Placebo
Sex					
Male	15 (48%)	14 (44%)	8 (28%)	χ^2^(1) = 2.74, *p* = 0.098^a^	χ^2^(1) = 1.72, *p* = 0.19^a^
Female	16 (52%)	18 (56%)	21 (72%)		
Age					
(years)	22.7 (1.4)	22.5 (1.1)	22.8 (1.4)	*t*(58) = −0.22, *p* = 0.83^b^	*t*(59) = −0.90, *p* = 0.37^b^
Body mass index					
(kg × m^−2^)	22.4 (2.6)	22.1 (2.3)	22.2 (2.5)	*t*(58) = 0.22, *p* = 0.83^b^	*t*(59) = −0.15, *p* = 0.88^b^
Smoking status					
Not at all	29 (94%)	29 (91%)	26 (90%)	*p* = 0.67^c^	*p* = 1.0^c^
Max. 5 cigarettes a day	2 (6%)	3 (9%)	3 (10%)		
Faculty					
of Medicine	22 (71%)	21 (66%)	20 (69%)	χ^2^(1) = 0.03, *p* = 0.87^a^	χ^2^(1) = 0.08, *p* = 0.78^a^
of Military Medicine	9 (29%)	11 (34%)	9 (31%)		
Perceived health status^d,e^					
From “very bad” (1) to “very good” (5)	4 (4–5)	4 (4–5)	4 (4–4)	*Z* = 1.81, *p* = 0.070^f^	*Z* = 1.24, *p* = 0.21^f^
Eating healthy diet^d,e^					
From “definitely not” (1) to “definitely yes” (5)	4 (3–4)	4 (3–4)	4 (3–4)	*Z* = 0.70, *p* = 0.49^f^	*Z* = 0.38, *p* = 0.70^f^
Consumption of fermented products^d,e^					
From “very rarely” (1) to “very often” (5)	3 (2–4)	3 (3–4)	3 (3–4)	*Z* = −0.91, *p* = 0.36^f^	*Z* = −0.38, *p* = 0.71^f^
Perceived economic status^d,e^					
From “very low” (1) to “very high” (5)	4 (4–5)	4 (4–4.5)	4 (4–5)	*Z* = 0.18, *p* = 0.85^f^	*Z* = 0.09, *p* = 0.93^f^
Psychometrics^e^					
EAT-26	8 (3–13)	7.5 (4.5–14)	9 (4–13)	*Z* = −0.19, *p* = 0.85^f^	*Z* = 0.06, *p* = 0.95^f^
BDI	6 (3–12)	5 (1.5–9)	8 (2–11.5)^g^	*Z* = −0.09, *p* = 0.98^f^	*Z* = *−*1,03, *p* = 0.30^f^
PSS-10	16 (10–25)	14 (11.5–20)	16 (12–23)	*Z* = 0.16, *p* = 0.88^f^	*Z* = −0.69, *p* = 0.49^f^
STAI trait	42 (36–49)	40 (33.5–46.5)	40 (33–47)	*Z* = 0.72, *p* = 0.47^f^	*Z* = *−*0.35, *p* = 0.72^f^
“Basal” outcome measures					
STAI state^e^	34 (30–40)	33 (27.5–40.5)	37 (31–43)	*Z* = −0.71, *p* = 0.48^f^	*Z* = −1.29, *p* = 0.20^f^
Salivary cortisol (ng/mL)^e^	1.71 (1.32–3.40)	2.58 (1.63–3.71)	2.36 (1.69–3.68)	*Z* = −1.24, *p* = 0.21^f^	*Z* = 0.31, *p* = 0.76^f^
Salivary metanephrine (pg/mL)	29.6 (15.6)	35.4 (17.5)	31.3 (18.2)	*t*(58) = −0.39, *p* = 0.70^b^	*t*(59) = 0.89, *p* = 0.38^b^
Pulse rate (min^−1^)	68.8 (9.5)	70.0 (8.9)	73.8 (11.4)	*t*(58) = −1.85, *p* = 0.070^b^	*t*(59) = −1.43, *p* = 0.16^b^

^a^ Pearson’s χ^2^ test, ^b^ Student’s *t*-test, ^c^ Fisher’s exact test, ^d^ rated on a single-item five-point Likert scale, ^e^ data reported as median (1st–3rd quartile), ^f^ asymptotic Mann Whitney *U* test, ^g^ one case with a missing value. EAT-26—Eating Attitudes Test-26, BDI—Beck Depression Inventory, PSS-10—Perceived Stress Scale-10, STAI—State-Trait Anxiety Inventory.

**Table 2 nutrients-12-01469-t002:** Intercorrelation of the outcome measures: Pearson’s *r* coefficients together with *p*-values are reported for the correlations of final examination scores and “basal” to “preexamination” increase in the rest of the measures. Statistically significant results are presented in bold.

	Increase in State Anxiety	Increase in Salivary Cortisol *	Increase in Salivary MN	Increase in Pulse Rate *
Examination score	0.10 *p* = 0.47	0.24 *p* = 0.087	0.09 *p* = 0.53	0.22 *p* = 0.12
Increase in state anxiety		0.14 *p* = 0.35	0.02 *p* = 0.91	**0.29 *p* = 0.040**
Increase in salivary cortisol			0.24 *p* = 0.091	−0.10 *p* = 0.48
Increase in salivary MN				**0.35 *p* = 0.012**

MN—metanephrine, * increase between the log-transformed variables, Kaiser-Meyer-Olkin measure = 0.371; the Bartlett’s test of sphericity: χ^2^(10) = 24.21, *p* = 0.0071. Color representation of the absolute values of Pearson’s correlation coefficients is as follows: no color: 0–0.1;  : 0.1–0.2;  : 0.2–0.3;  : 0.3–0.4.

**Table 3 nutrients-12-01469-t003:** Evaluation of the tested product by participants: At the study completion, the participants were asked a question “which product, in your opinion, *Saccharomyces*, *Lactobacillus*, or placebo, did you take?” The products indicated by them were compared with the actually taken.

		Product Indicated by Participants			Total
		S	L	P	
Product actually taken	S	6 (24%)	5 (20%)	14 (56%)	25 (100%)
	L	4 (15%)	13 (50%)	9 (35%)	26 (100%)
	P	7 (27%)	8 (31%)	11 (42%)	26 (100%)
Total		17	26	34	77

S—*Saccharomyces*, L—*Lactobacillus*, and P—placebo. No significant link was found between the actual distribution of taken products and participant’s opinion on which product did they take: χ^2^(4) = 5.70, *p* = 0.22

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
