# Peer review of "Effect of Supplementation with Saccharomyces Boulardii on Academic Examination Performance and Related Stress in Healthy Medical Students: A Randomized, Double-Blind, Placebo-Controlled Trial"

_nutrients, 2020, doi:10.3390/nu12051469_

Round 1

Reviewer 1 Report

Firstly, I hope the authors and editors are all keeping well in this unprecedented time!

In this manuscript (“Effect of supplementation with Saccharomyces boulardii on examination-related stress in healthy medical students: a randomized, double-blind, placebo-controlled trial”), Karbownik and colleagues examined the effect of a yeast probiotic on academic performance as well as stress-related anxiety and biomarkers. The study was largely negative, finding no effect of the probiotic on academic grades, anxiety levels, cortisol or metanephrine. The probiotic did increase stress-induced pulse rate rises, which to me seems (at least weakly) indicative of poorer response to exam stress in the treatment group. While I do not think that these negative findings should hamper publication of this work (in fact, I think it is vital to publish negative studies in this field that is often criticized as over-hyped), my main concern is that the manuscript in its current form seems to promote the potential of the probiotic as a treatment for stress. A more balanced approach that better reflects the study findings must be taken for the manuscript to become suitable for publication. In addition, I have outlined some other comments and queries about each section for the authors below.

Introduction

The introduction was generally concise and clear. It would be useful to add some background information on metanephrine and its relevance to stress and the current study (either in the introduction or methods). Given the pattern of results, metanephrine was one of the main points in the discussion section, yet it was not clear why it was chosen as an outcome measure in the first place.

Materials and Methods

Regarding the Lactobacillus product, it would be beneficial to include a little more information on what is meant by “degraded” in the main manuscript as it raises a lot of questions in the reader’s mind. Indeed, it would be good to have a summary of the quality control results in general in the main manuscript (including the actual CFU per dose).

The description of the results of the pre-test on lines 160-162 is unclear; is 38% the total for each of the “frightening” and “considerable” categories separately or combined? Either way, the current phrasing suggests that there were only 3 response options but then the percentages don’t add up.

The phrasing for each phase of the study could be clarified. Currently, the authors use “pre-test” and “pre-examination test”, which are easily confused. Also, “pre-examination” seems to refer to 2 different testing sessions on different days?

To aid with this, the figure could also be clarified and moved earlier in the methods section. In the figure, these are the points that were unclear for me:

  • What does “Pre-study” refer to? I think this can probably be removed from the timeline.
  • How long is the pre-intervention period in days? The arrows seem to indicate that enrolment, allocation, and basal salivary sample collection all happen on separate days – is this correct?
  • Same query for the information under “Intervention Day 30”: the arrows imply that pre-examination, final examination, and post-examination were on different days, even though the heading and the in-text description both indicate otherwise.

Some of the methods should be expanded:

  • Who conducted the medical interview? Was this done one-on-one?
  • Why was only vegan diet identified and excluded?
  • What counted as chronic medication? (e.g. did the contraceptive pill, asthma treatments or other “minor” medications lead to exclusion?)
  • The strategy of asking participants to allocate their own participant codes seems a bit risky. Did you check for possible errors, and if so, how was this done without breaking blind?
  • I understand that these are medical students, but it still seems a bit strange to rely solely on self-reported pulse measurements
  • How was saliva collected? With salivette/gauze or straight in a collection tube?
  • Why was there a focus on consumption of fermented products but not other dietary components?
  • The “other general linear models” description on line 306 is too vague – what were factors of interest and how were they selected?

Results

It was nice that both an ITT and per protocol analysis were conducted with similar results.

The information on dropouts was confusing, mostly because it was unclear when the Lactobacillus group was included or not.

  • On line 287 of the methods, it indicated n=6 dropouts from S. boulardii group and n=4 dropouts from the placebo group, but in the results (lines 316-317, 321-323) lots of other numbers are thrown around and only when I looked at the figure did it become clear that some of these were from the other group not included in any other place in the manuscript.

On line 320, what does it mean that participants were “followed up for 30 days”? It could imply that participants were contacted each day during this period, but I don’t think that is the case?

For the sociodemographic and psychometric variables, the actual statistics should also be reported somewhere in addition to the p values.

Discussion and conclusion

As noted in my initial comments, I found that the discussion focused too much on the single significant finding of the study (pulse rate). I found it to be overly long and very speculative, particularly in the discussion of the mechanistic link between pulse rate and metanephrine (and especially since there was no observed difference in metanephrine levels between groups in the present study). The conclusions were rather vague.

Given that there was only one significant finding, it is worth considering that this was simply a Type I error. I think it is particularly problematic to focus so heavily on this one result when it was based upon changes in self-measured pulse rate, with no indication of the accuracy of this measurement or how accuracy might be affected by stress.

If this study is to be published, both the discussion and conclusion must be updated to reflect that the vast majority of outcomes examined were not found to be significantly altered by the intervention.

Other minor comments

  • Line 62: this sentence was difficult to understand, could be rephrased more clearly
  • Line 65: sentence could be made clearer by moving “tested as anti-anxiety agents” to be after “vast majority of probiotic species”
  • Line 226: there is a typo (“measurands” should be “measurements”)
  • Line 291: missing a word in “some the other”
  • Line 307: “with 2 significant figures” is confusing in a discussion of statistics
  • Line 323: “enormously high” does not sound very scientific
  • Line 325: “17%, of drop-out rate” should be just “17% drop-out” – should also explicitly state that this refers to the whole study
  • In tables, * conventionally indicate significance. Please use different symbols to indicate the different footnotes
  • Line 517: “a case” should be “the case”

Reviewer 2 Report

This study of dietary supplementation with a yeast probiotic to reduce stress and boost cognitive function among medical students during examinations is timely and important. Although the findings are not positive, it is an important study in that it helps gauge the potential of the supplement and its autonomic side effects. The study appears carefully designed and mostly well conducted. There are a few issues, in particular with the presentation of the study, which require the authors’ attention.

  1. The framing of the study highlights the role of stress in examinations, however, the primary outcome variable is performance-related, which does not necessarily inform about stress. The critical variables of stress are well-devised on experiential, autonomic, and endocrine levels, however, they are only ranked secondary in importance. This appears somewhat inconsequential and the authors should try to address that by either providing a stronger rationale for testing exam performance or changing the priority of results presented.
  2. Although the data analysis paragraph outlines ANOVA designs, I do not see critical Group by Time interactions reported. Rather, simple means comparisons within and between groups are provided.
  3. Metanephrine shows considerable spread. Was normality established or would actually transformations (e.g. natural logarithm) be needed? This would also be important before calculating Pearson correlation coefficients across study groups.
  4. Table one is cumbersome and drowns in superfluous detail. “Nationality Polish” does not need to be reported if it is 100% in all groups. The ad-hoc scales for perceived health status to economic status could also be shown as means and analyzed continuously. The format of the table (and the rest of the manuscript) should adhere to journal standards.
  5. More details should be provided on how the study was framed, what potential expectations participants had regarding effects of the supplement(s).
  6. a limitation is the failure to track intake (e.g. by electronic pill boxes). This type of adherence tracking is superior although not yet implemented widely.
  7. Grammar and style would profit from additional native speaker correction.

Round 2

Reviewer 1 Report

The authors have addressed all of my previous comments satisfactorily and I am happy to recommend publication. I have only one very minor suggestion to define “overuse” with respect to alcohol in the exclusion criteria (as I expect that alcohol use would be quite common amongst university students).

Author Response

Dear Reviewer,

Thank you for your positive feedback. As you requested, we have defined “alcohol overuse” according to the Polish standards.

Kind regards,

Michał Karbownik

Reviewer 2 Report

The authors have addressed my concerns, although I still find the focus of performance as primary outcome less convincing, because according to the rationale developed in the introduction, the primary aim of the intervention would have been to reduce stress, which would then have mediated an improvement in performance.

A minor issue of the presentation is that sometimes there is no consistency in past tense.